# PRISM: High-Resolution & Precise Counterfactual Medical Image Generation using Language-guided Stable Diffusion

**Amar Kumar**[1,2]                                              AMAR.KUMAR@MAIL.MCGILL.CA

**Anita Kriz**[1,2]                                              ANITA.KRIZ@MAIL.MCGILL.CA

**Mohammad Havaei**[3]                                          MHAVAEI@GOOGLE.COM

**Tal Arbel**[1,2]                                              TAL.ARBEL@MCGILL.CA

[1] *Center for Intelligent Machines, McGill University, Montreal, Canada.*

[2] *Mila - Quebec AI Institute, Montreal, Canada.*

[3] *Google Research, Montreal, Canada.*

**Editors:** Accepted for publication at MIDL 2025

## Abstract

Developing reliable and generalizable deep learning systems for medical imaging faces significant obstacles due to spurious correlations, data imbalances, and limited text annotations in datasets. Addressing these challenges requires architectures that are robust to the unique complexities posed by medical imaging data. Rapid advancements in vision-language foundation models within the natural image domain prompt the question of how they can be adapted for medical imaging tasks. In this work, we present PRISM, a framework that leverages foundation models to generate high-resolution, language-guided medical image counterfactuals using Stable Diffusion. Our approach demonstrates unprecedented precision in selectively modifying spurious correlations (the medical devices) and disease features, enabling the removal and addition of specific attributes while preserving other image characteristics. Through extensive evaluation, we show how PRISM advances counterfactual generation and enables the development of more robust downstream classifiers for clinically deployable solutions. To facilitate broader adoption and research, we make our code publicly available at https://github.com/Amarkr1/PRISM.

**Keywords:** Counterfactual Image Synthesis, Diffusion, Foundation Models, Generative Models, Large Language Models

## 1. Introduction

The development of deep learning models in healthcare settings has the potential to transform current medical practices in disease diagnosis, biomarker discovery, and personalized treatment. However, clinical deployment requires robust models – a standard that remains largely unmet due to the inherent complexities of medical imaging data. Class imbalances and spurious correlations can cause models to learn misleading patterns that are not penalized when optimizing the training objective. This flawed training paradigm results in incorrect disease classification, ultimately degrading the model's generalizability to real-world clinical scenarios. To address these challenges, the field has explored counterfactual (CF) generation to expose shortcut learning and alleviate data imbalance issues by augmenting underrepresented classes. Previous work has focused on classifier-guided counterfactual image generation methods, such as using standard classifiers with robust

empirical minimization techniques (Mertes et al., 2022; Singla et al., 2019) or classifiers based on distributional robust optimization (Group-DRO) (Kumar et al., 2023; Fathi et al., 2024). An alternative approach leverages Structural Causal Models (SCMs) to explicitly model and intervene on causal relationships between attributes during the generation process; these methods also (largely) rely on classifiers to produce high-quality results (Ribeiro et al., 2023). These methods expose a paradox in their formulation – their performance is dependent on the same biased data (and classifiers) they are designed to mitigate (see Fig. 1). Moreover, end-to-end architectures face a tradeoff between competing objectives: high-quality generation demands fine-grained details, while classification relies on abstract features. Compounded by the computational burden of training high-capacity architectures from scratch, synthesizing high-resolution and precise CFs remains elusive.

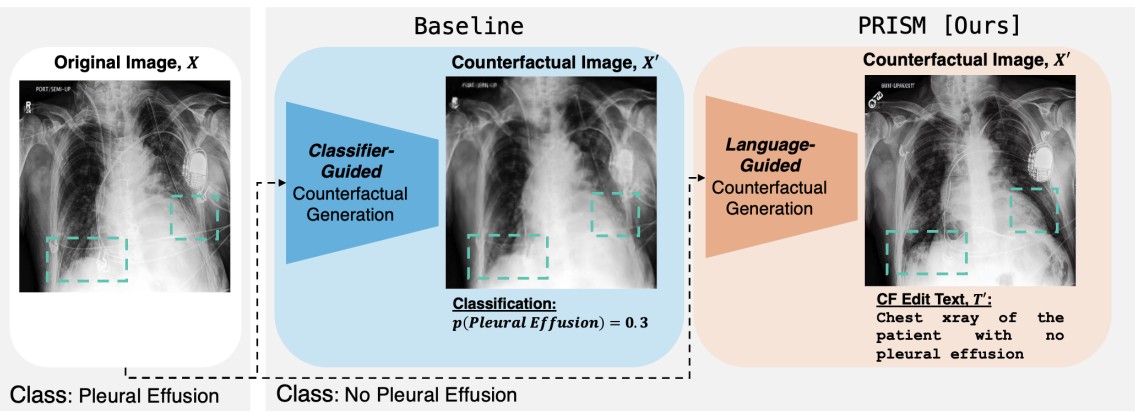

Figure 1: Counterfactual (CF) explanations for a subject with 'Pleural Effusion'. (a) Original chest radiograph of subject; (b) Classifier-guided CF image fails to show changes in the diseased area and determined the CF image is healthy. The classifier is biased and associates the disease with the medical device; (c) PRISM modifies the area of disease pathology, leaving the devices (e.g. pacemaker) unchanged.

Fine-tuning foundation models has recently emerged at the forefront of deep learning for medical image analysis (Wang et al., 2023; Dutt et al., 2023; Azad et al., 2023), outperforming existing state-of-the-art (SOTA) methods in tasks such as zero-shot classification (Yuan et al., 2021), out-of-distribution generalization (Goyal et al., 2023), histopathology image classification (Roth et al., 2024), and visual question answering (Li et al., 2024). In computer vision, many methods have been developed for high-resolution, language-guided image editing (e.g.Null-text Inversion (Mokady et al., 2023), Imagic (Kawar et al., 2023)). BiomedJourney (Gu et al., 2023) was the first work to fine-tune foundation models for counterfactual medical image generation via language prompts and achieved SOTA results. However, it was not designed to remove large confounding artifacts (e.g. medical devices) and is constrained to low resolution images ($256 \times 256$). RadEdit (Pérez-García et al., 2025) employs language-guided image editing to address biases from acquisition, manifestation, and population shifts. It uses two masks: one to define areas where edits can occur and another to maintain fidelity. This limits its ability to generate fully unconstrained counterfactuals. This raises a natural question: *Could we leverage a vision-language foundation*

*model pre-trained on diverse natural images and adapt it to generate precise high-resolution medical image counterfactuals?*

In this work, we introduce PRISM (**Pr**ecise counterfactual **I**mage generation using language-guided **S**table Diffusion **M**odel), a strategically fine-tuned vision-language foundation model, that leverages language guidance to generate medical image counterfactuals for novel generative tasks (see Fig. 2). Specifically, PRISM presents the first framework to generate high-resolution ($512 \times 512$) medical counterfactuals that can selectively remove significant spurious artifacts, such as medical devices. Crucial for explainability in medical settings, it can isolate and modify individual disease attributes (and spurious correlations) while preserving others. Existing approaches have relied on detailed clinician notes to train language models (Zhang et al., 2023; Luo et al., 2024). In order to leverage the guidance of a language embedding, our framework adapts binary labels, typical for medical datasets, into text captions.

Through extensive experimentation on the publicly available CheXpert dataset (Irvin et al., 2019), we validate our approach by (i) generating difference maps between the original and the synthesized CF image to assess the clinical plausibility of the disease, and (ii) using multi-head classifiers to confirm that the counterfactuals are correctly classified. We also show improvement over a baseline classifier-guided GAN-based model, GANterfactual (Mertes et al., 2022). As a key demonstration of PRISM's utility, we show that our counterfactuals improve the accuracy of an existing classifier.

## 2. Methodology

While state-of-the-art vision-language foundation models in computer vision utilize millions of image-text pairs to generate images, their direct application to the medical domain is hindered by two key challenges. First, patient information is stored as tabular data (e.g., numerical labels for age or sex) rather than descriptive text , limiting direct integration into existing vision-language models. Second, medical imaging datasets are significantly smaller than those in computer vision, making it impractical to train a foundation model from scratch. To address these shortcomings and enable CF generation, our methodology consists of three main steps: (i) convert patient tabular data into text format, enabling the generation of rich semantic embeddings via a pre-trained CLIP (Contrastive Language-Image Pre-training) text encoder, Section 2.1; (ii) fine-tune a Stable Diffusion model, to better adapt to a medical imaging dataset, Section 2.2; (iii) at inference, synthesize CF images guided by a text input, Section 2.3.

### 2.1. Tabular Data to Text Conversion

One of the key requirements of training a Stable Diffusion (specifically v1.5) (Rombach et al., 2022) model is the image-text pair. CheXpert, the medical dataset we use here, only contains binary labels for different diseases and the presence of support devices. To leverage Stable Diffusion, we create a custom template for image-text pairs based on the available tabular data (see code listing in Appendix A). For example, if the subject's radiograph shows pleural effusion and cardiomegaly, our text caption for the image is `chest x-ray of a patient showing pleural effusion, cardiomegaly`. Additionally, for patients

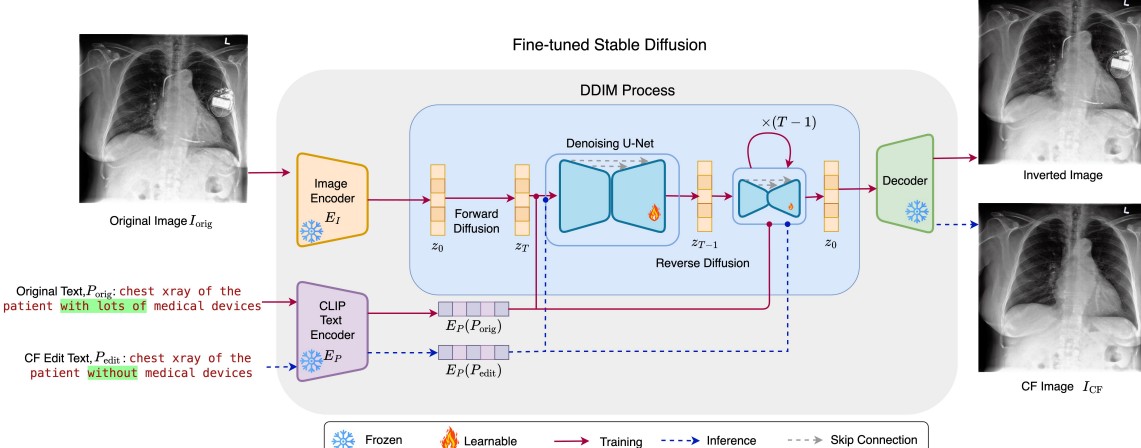

Figure 2: The framework for synthesizing counterfactual (CF) images takes an original input image ($I_{\text{orig}}$) and its corresponding text prompt ($P_{\text{orig}}$), along with an edited text prompt ($P_{\text{edit}}$) for the CF image. It employs a frozen VAE consisting of an image encoder ($E_I$) and decoder, as well as a frozen CLIP text encoder ($E_P$). The core component of the framework is a denoising U-Net, which is the only trainable module during the fine-tuning process. During inference, the encoded text prompt ($E_P(P_{\text{edit}})$) is used to condition the U-Net, guiding the generation of a high-resolution $512 \times 512$ counterfactual image that aligns with the modified text description.

with no findings, we use the template text `Normal chest X-ray with no significant findings`.

## 2.2. Fine-Tuning the Stable Diffusion Model

The Stable Diffusion v1.5 architecture consists of three components: (i) the Variational Autoencoder (VAE), which encodes images into the latent space and subsequently decodes the processed latent representation back into image space; (ii) the U-Net, which operates at the latent level and is trained to predict and remove noise introduced during the forward diffusion process, enabling iterative image refinement; and (iii) the CLIP Encoder, which encodes text descriptions into a vector embedding that is used to condition the U-Net, guiding the image generation process to match the given text description. It should also be noted here that the CLIP model is already pre-trained, providing general semantic knowledge about image-text relationships.

The conditional U-Net architecture learns to predict noise based on noisy latents (noise image embeddings), timesteps (indicating noise level) and text embeddings (embeddings from the CLIP text encoder). A Mean Squared Error (MSE) loss is computed between the predicted and the actual noise. Finally, the backpropagation updates only the U-Net weights, excluding the VAEs. We provide full pseudocode and details for implementing the fine-tuning process in Appendix A.

## 2.3. Generating Counterfactuals at inference

After fine-tuning the Stable Diffusion model on a medical imaging dataset, generating counterfactuals requires no extra training and is done at inference. To generate identity-preserving counterfactuals, we adopt the image-editing component of LANCE (Prabhu et al., 2023), which combines DDIM with null-text inversion for precise image editing that maintains fidelity to the original image. The three main steps in CF generation include image inversion, image editing and image quality evaluation. Additional details for image editing are discussed in Appendix A.1. To produce a precise counterfactual image ($I_{\mathrm{CF}}$), the language embeddings of the CF edit text ($P_{\mathrm{edit}}$) are used as contexts within the U-Net to guide the denoising process applied to the diffused latent representation of the input image ($I_{\mathrm{orig}}$). The text embeddings are incorporated into the denoising U-Net during the reverse diffusion process using cross-attention modules. To quantify the alignment of the counterfactual image with the provided edited text alignment, we use an editing score, $S_{\mathrm{CLIP}}$ (Eq. 1), which measures the similarity between the generated image and the intended textual modification. Following a similar approach to (Prabhu et al., 2023), we compute the editing score and directional similarity (Gal et al., 2022) to filter out edited samples where $S_{\mathrm{CLIP}} < 0.1$. All details required to perform language-guided image editing are discussed in Appendix A.1.

$$S_{\mathrm{CLIP}} = \frac{\Delta I \cdot \Delta P}{\|\Delta I\|\|\Delta P\|}, \quad \text{where} \quad \begin{aligned} \Delta I &= E_I\left(I_{\mathrm{CF}}\right) - E_I\left(I_{\mathrm{orig}}\right), \text{ and} \\ \Delta P &= E_P\left(P_{\mathrm{edit}}\right) - E_P\left(P_{\mathrm{orig}}\right) \end{aligned} \tag{1}$$

## 3. Experiments and Results

### 3.1. Dataset and Implementation Details

We use the publicly available CheXpert dataset (Irvin et al., 2019) that contains over 200,000 chest X-ray images, with binary labels for 14 diseases including the presence of support devices. Table 1 shows a summary of the number of subjects in each split and their distributions. To demonstrate our method's versatility to other medical datasets, we additionally ran experiments on dermoscopic images from the publicly available ISIC dataset (Tschandl et al., 2018; Codella et al., 2018; Combalia et al., 2019), with results and details discussed in Appendix C.

PRISM uses the default DDPM scheduler for fine-tuning the model 'runwayml/stable-diffusion-v1-5'. There is a known tradeoff between a lower noise scheduler and diversity in the sampled results. Choosing a lower noise scheduler (12e-3 in this case) tends to produce a more detailed image with less noise and generates deterministic results. Additionally, the convergence is faster, as only a few sampling steps are required (Song et al., 2020). The implications of this tradeoff should be explored in each context of interest. During image editing (CF synthesis), we utilize a DDIM scheduler with a `scaled_linear` scheduler with `beta_start` and `beta_end` as `85e-5` and `12e-3` respectively. These parameters define the range of noise variance ($\beta$) added at each timestep and linearly increase from `beta_start` to `beta_end`. Text similarity is computed based on `cosine_similarity`. Additionally, for all the synthesized counterfactual images discussed in this manuscript, we use the same hyper-parameters (e.g. denoising steps, DDIM scheduler) for all tasks, except the language-based

command for each case. Thus, our proposed method does not need extensive hyperparameter tuning. We provide additional implementation details in Appendix A and the code along with model weights for the fine-tuned Stable Diffusion are publicly available at https://huggingface.co/amar-kr/PRISM.

| Attribute → Splits↓ | Pleural Effusion | Cardiomegaly | No Finding | Support Devices |
|---|---|---|---|---|
| Train | 62509 | 21888 | 12222 | 78211 |
| Validation | 10996 | 3739 | 2161 | 13678 |
| Test | 12972 | 4515 | 2591 | 16196 |

Table 1: Summary of the number of samples for train, validation and test splits.

### 3.2. Experiments and Metrics: Evaluating the Generated CF Images

To establish baseline comparisons, we implement GANterfactual (Mertes et al., 2022), a classifier-guided CF image generation method. We fine-tune pre-trained Efficient-Net (Tan and Le, 2019), initially trained on Image-Net, for a multi-head classification task: pleural effusion, cardiomegaly, no finding and support devices. This classifier is then used to verify the class of the CF images synthesized by our PRISM framework, ensuring that the generated CFs accurately reflect the desired modifications of the correct target class. It should be noted that the baseline method requires an image size of $224 \times 224$.

To quantitatively evaluate the quality of synthesized counterfactual images, we use the following metrics: (i) **Subject Identity Preservation** evaluates how well the subject-identifying characteristics are maintained while only modifying the targeted attribute. Following prior work (Mothilal et al., 2020; Nemirovsky et al., 2020), this is calculated through the $L_1$ distance between the CF and factual images. (ii) **Counterfactual Prediction Gain (CPG)** (Nemirovsky et al., 2020) measures the absolute difference in a classifier's predictions between factual and CF images. A higher CPG indicates a greater shift across the classifier's decision boundary. To this end, we trained a binary AlexNet (Krizhevsky et al., 2012) to detect the presence (1) or absence (0) of medical support devices (e.g. pacemakers, wires, tubes) in the original images. Then at inference, this AlexNet model measures the CPG score for the CF images synthesized by PRISM and the baseline method, respectively.

A final set of experiments is devised in order to show that the synthesized CF images focus on the defining features of each disease (such as pleural effusion occurring at the corner of the lungs or cardiomegaly surrounding the position of the heart). The training data for the original EfficientNet classifier is then augmented with these CF images. Each subgroup - Pleural Effusion, Cardiomegaly, No Findings and Support Devices are augmented with 2500 CF samples. An increase in classifier accuracy suggests that synthesized counterfactual images enhance generalizability and robustness, enabling the classifier to identify defining disease features independent of potential confounding factors in the dataset. This is particularly important in the context of pleural effusion, which is correlated with the presence of medical devices. To validate the hypothesis that CF image augmentation enhances subgroup-level performance compared to generic augmentation, we perform a controlled

|                                          | L1↓       | CPG↑      |
|------------------------------------------|-----------|-----------|
| Baseline (classifier-guided GAN-based CF) | 0.091     | 0.781     |
| PRISM [**Ours**]                         | **0.031** | **0.845** |

Table 2: Quantitative results comparing the scores for the CF generated by a classifier-guided GAN-based method and PRISM, when asked to remove the medical devices. The high CPG indicates that PRISM synthesizes CF images that correctly change the class labels.

experiment. In this setup, each subgroup is augmented with 2500 samples generated from the fine-tuned Stable Diffusion model with the following text prompts: `Chest X-ray of a patient with severe cardiomegaly and without support devices`, `Chest X-ray of a patient with no findings`, `Chest X-ray of a patient with pleural effusion and lots of  support devices`, and `Chest X-ray of a patient with pleural effusion and without support devices`.

### 3.3. Results

**Classifiers** EfficientNet has a classification accuracy of 0.8, 0.87, 0.91 and 0.86 for pleural effusion, cardiomegaly, no finding and support devices, respectively (see first row of Table 3). The accuracy and AUC of the binary AlexNet classifier on a held-out test set are 0.89 and 0.91, respectively. These classifiers are used to measure the CPG scores reported in Table 2.

**Qualitative Evaluations** Our qualitative evaluation demonstrates two primary capabilities of our method: (i) the ability to remove and, for completeness, *add* medical devices to the original image, and (ii) the ability to emulate distinct visual pathologies of different diseases.

Chest radiographs contain a variety of medical devices (Gambato et al., 2023) such as chest tubes for draining air, blood, or fluid from the pleural space, surgical clips that are often visible after procedures like axillary node dissection, or pacemakers that regulate heart rhythm, typically seen as a small box near the clavicle (Mathew et al., 2019). These devices vary in shape, size and position in the X-ray image. Our method, PRISM, can remove medical devices, demonstrating robust performance across various device types and positions without any external classifier-based supervision or image-level mask/annotations. In Fig. 3, we show how, by using language guidance, we can remove complex medical devices from the given image without altering the pathology of the disease. We also compare our framework to a baseline method, GANterfactual (Mertes et al., 2022), a classifier-guided CF generator. This method relies on the gradient from a pre-trained classifier for guidance and fails to remove devices from the image. Next, we evaluate our method's ability to effectively *differentiate* between diseases during CF image generation. Specifically, Fig. 4 demonstrates PRISM's performance in generating CFs for two diseases: Pleural Effusion and Cardiomegaly. The difference maps in Fig. 4 demonstrate that our approach can identify and remove the target disease while preserving the anatomical features of the subject, as well as the devices and other artifacts outside the regions of the expected changes. In Appendix B, we provide additional qualitative comparisons between PRISM and state-of-

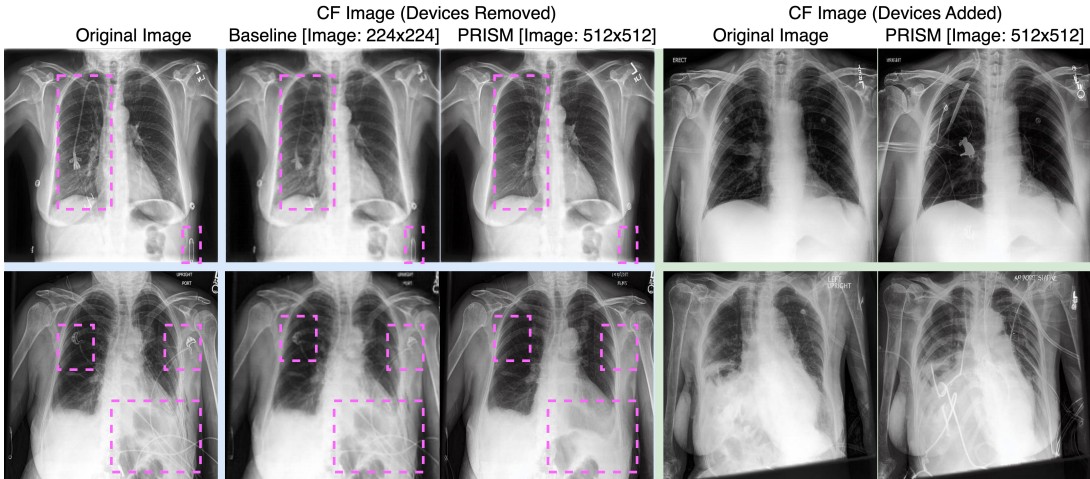

Figure 3: Sample pairs of original and CF images demonstrate the capability of PRISM to remove and add medical devices (e.g. wires, pacemaker) in high resolution. Left: CF images with medical devices removed. Language guidance is $T$: `chest xray of the patient with lots of medical devices`, $T'$: `chest xray of the patient without medical devices`. Note that the baseline method cannot properly remove medical devices; Right: CF images with added medical devices. Language guidance is $T$: `chest xray of the patient with no support devices`, $T'$: `chest xray of the patient with lots of support devices`.

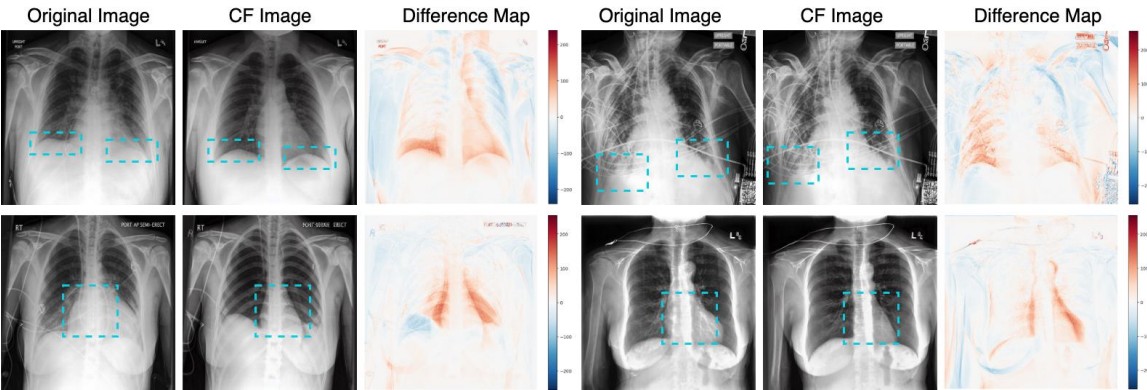

Figure 4: Sample pairs of original and edited images showcasing accurate, precise and high-resolution generated CFs for disease pathology explainability. The original ($T$) and edited text prompts ($T'$) are - Row 1: $T$ - `chest x-ray of the patient with severe pleural effusion`, $T'$ - `chest x-ray of the patient with no finding`; Row 2: $T$ - `chest x-ray of the patient with severe cardiomegaly`, $T'$ - `chest x-ray of the patient with no finding`.

the-art (SOTA) text-guided image editing methods, including Imagic (Kawar et al., 2023), Null-text Inversion (Mokady et al., 2023), and RadEdit (Pérez-García et al., 2025). The

results demonstrate PRISM's ability to generate precise CF images that remain consistent with the original factual image, outperforming other methods.

**Quantitative Evaluations** To quantitatively evaluate our approach, we compare our method with GANterfactual, a classifier-guided GAN-based approach for generating counterfactuals. Table 2 shows the results for the task of removing medical devices. The counterfactual images generated by GANterfactual show similar $L_1$ scores to those produced by our method, indicating that the synthesized images in both cases remain close to their factual counterparts. However, counterfactuals generated by PRISM achieve higher CPG scores, suggesting that these images are more effectively converted to the opposite class (see Appendix F for additional results).

Table 3 shows the results of re-training the classifier with CFs for the classes Pleural Effusion, Cardiomegaly, No Finding, and Support Devices. As shown, augmented training leads to improved classifier performance, demonstrating that incorporating CFs synthesized by PRISM enhances the model's robustness. Notably, this increase in performance is not observed when the original data is randomly augmented with samples from the fine-tuned stable diffusion model, thus supporting the hypothesis that CF augmentation specifically improves classifier performance.

|  | Pleural Effusion | Cardiomegaly | No Finding | Support Devices |
|---|---|---|---|---|
| Original Data | 0.80 | 0.87 | 0.91 | 0.86 |
| Original Data + SD samples | 0.82 | 0.86 | 0.91 | 0.85 |
| Original Data + PRISM CFs | **0.88** | **0.90** | **0.92** | **0.88** |

Table 3: Augmented classifier accuracies using Efficient-Net: Synthetic samples from PRISM [second row - Original Data+SD (Stable Diffusion) samples] and CF images generated by PRISM [third row - Original Data + PRISM CFs] are used to augment the training dataset. The accuracies are reported on the same held-out test set.

## 4. Conclusion

Developing a generative model in the medical domain to produce high-quality counterfactuals requires a balance between image fidelity and controllability. In this work, we present PRISM, a fine-tuned vision-language foundation model for counterfactual medical image generation that addresses these challenges. PRISM is the first framework to use language guidance to synthesize high-resolution ($512 \times 512$) medical images consistent with their factual counterparts. We demonstrate our results through extensive experiments on the CheXpert dataset. Our approach generates precise and accurate CFs representing disease states and is able to cleanly remove medical devices. We make our code and fine-tuned model weights publicly available to the medical imaging community for further development. Future work will investigate the use of synthesized counterfactual images to build robust classifiers for out-of-distribution generalization, and to assess the disentanglement capacity of language-guided foundation models.

## Acknowledgements

The authors are grateful for funding provided by the Natural Sciences and Engineering Research Council of Canada, the Canadian Institute for Advanced Research (CIFAR) Artificial Intelligence Chairs program, Mila - Quebec AI Institute, Google Research, Calcul Quebec, Fonds de recherche du Québec (FRQNT), and the Digital Research Alliance of Canada.

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

# Appendix A. Additional Implementation Details and Code Listings

We provide additional steps for our implementation.

---

**Code Listing 1** Generating text for the images in CheXpert dataset

```python
1   conditions = [
2               'No Finding', 'Enlarged Cardiomediastinum', 'Cardiomegaly',
3               'Lung Opacity', 'Lung Lesion', 'Edema', 'Consolidation',
4               'Pneumonia', 'Atelectasis', 'Pneumothorax', 'Pleural Effusion',
5               'Pleural Other', 'Fracture', 'Support Devices'
6           ]
7
8   captions = []
9   for image in images:
10      findings = []
11      for condition in conditions:
12          if image[condition] == 1:
13              findings.append(condition)
14
15      caption = "Chest X-ray showing " + ", ".join(findings) if findings else
        ↪  "Normal chest X-ray with no significant findings"
16      captions.append(caption)
```

---

**Algorithm 1:** Fine-tuning Stable Diffusion on CheXpert

**Pre-trained Stable Diffusion model components**: unet, vae, textEncoder, tokenizer, noiseScheduler

**CheXpert dataset**: dataloader

**Optimizer**: optimizer **for each** *batch in dataloader* **do**

  latents = vae.encode(batch["image"])          ▷ encode images into latent space
  noise = sampleRandomNoise()                   ▷ add random noise to latents
  timesteps = sampleRandomTimesteps()
  noisyLatents = noiseScheduler.addNoise(latents, noise, timesteps)
  encoderHiddenStates = textEncoder(batch["inputIds"])    ▷ encode text captions
  noisePred = unet(noisyLatents, timesteps, encoderHiddenStates)    ▷ predict noise residual with U-net
  loss = mseLoss(noisePred, noise)              ▷ compute pixel wise loss
  backward(loss)                                ▷ backpropagate
  optimizer.step()                              ▷ update weights
  optimizer.zeroGrad()

**end**

---

## A.1. Language-guided Image Editing using PRISM

To generate CF medical images with language guidance, PRISM adopts the image-editing technique used in LANCE (Prabhu et al., 2023), which combines Null-text inversion (Mokady et al., 2023) with Prompt-to-prompt (Hertz et al., 2022) attention manipulation. Algorithm 2 presents the detailed pseudo-code outlining the three key steps involved in PRISM's image editing process: (i) image inversion, (ii) image editing, and (iii) quality evaluation of the generated image.

**Image Inversion:** In the inversion stage, the objective is to recover a latent representation of the original image and optimize unconditional embeddings to ensure accurate reconstruction. First, the original image $I_{\text{orig}}$ is encoded into the latent space as $z_T$ using an image encoder $E_I$. A deterministic DDIM reverse diffusion then produces the latent sequence $\{z_T, z_{T-1}, \ldots, z_0\}$.

Unconditional embeddings $E_{\text{uncond}}$ are randomly initialized, while conditional embeddings $E_{\text{cond}}$ are derived from the original prompt $P_{\text{orig}}$. For each diffusion step (from $t = T$ to $t = 1$), a predicted latent $\hat{z}_{t-1}$ is computed using $E_{\text{cond}}$ and the current $E_{\text{uncond}}$. The mean squared error, $\mathcal{L} = \|\hat{z}_{t-1} - z_{t-1}\|^2$, is minimized via gradient descent to update $E_{\text{uncond}}$. This null-text inversion process aligns the latent representation with the original image, preserving its structure and style for accurate reconstruction and reliable editing. Figure 5 shows the original and inverted images, with many details preserved during generation. Notably, the model struggles with the small text found within the images, which we further discuss in Appendix I. When the original and inverted images are passed through the state-of-the-art classifier, the changes in multi-class logit values are close to zero. This confirms that the inversion process maintains relevant details needed for accurate image classification.

**Image Editing:** In this step, the model modifies the original image by initiating the denoising diffusion process from the latent representation $z_T$ obtained during the inversion step. The goal is to progressively refine this latent representation towards a clean, edited image while applying changes specified by the edited prompt.

The process begins by encoding the original prompt $P_{\text{orig}}$ and the edited prompt $P_{\text{edit}}$ into their respective conditional embeddings $E_{\text{cond}}^{\text{orig}}$ and $E_{\text{cond}}^{\text{edit}}$. For each timestep $t$ (from 1 to $T$), the model retrieves attention maps for both the original and edited prompts, $A_{\text{orig}}$ and $A_{\text{edit}}$, based on the current latent representation $z'_{t-1}$. Here, cross-attention is implemented similar to Prompt-to-prompt (Hertz et al., 2022). Once the diffusion process is completed, the final counterfactual image $I_{\text{CF}}$ is decoded from the final latent representation $z'_T$.

**Quality Evaluation** Once the image has been generated, the CLIP similarity score, $S_{\text{CLIP}}$ (as defined in Equation 1), is used to assess the quality of the edits. This score evaluates the similarity between the generated and original images and the alignment of the image with the edited text prompt (Prabhu et al., 2023).

---

**Algorithm 2:** Counterfactual Medical Image Generation using PRISM

---

**Input:** $I_{\text{orig}}$ (Original Image), $P_{\text{orig}}$ (Original Image Prompt), $P_{\text{edit}}$ (Edit Prompt), $E_I$ (Image Encoder), $E_P$ (Text Prompt Encoder), $f_\theta$ (Diffusion model)

**Output:** $I_{\text{CF}}$ (Counterfactual Image)

**Step 1: Image Inversion**

Encode image to latent space: $z_T \leftarrow E_I(I_{\text{orig}})$

Perform DDIM reverse diffusion to get latent sequence: $\{z_T, z_{T-1}, \ldots, z_0\}$

$E_{\text{uncond}} \leftarrow \text{Random\_Initialize}()$

$E_{\text{cond}} \leftarrow E_P(P_{\text{orig}})$

  **for** $t = T$ *to* 1 **do**

       $\hat{z}_{t-1} \leftarrow \text{DDIM\_Reverse\_Step}(z_t, E_{\text{cond}}, E_{\text{uncond}})$

       $\mathcal{L} \leftarrow \|\hat{z}_{t-1} - z_{t-1}\|_2^2$                ▷ MSE loss

       Update $E_{\text{uncond}}$ via gradient descent to minimize $\mathcal{L}$

  **end**

**Step 2: Image Editing**

  $z'_0 \leftarrow z_0$                             ▷ initialize with inverted latent

  $E_{\text{cond}}^{\text{orig}} \leftarrow E_P(P_{\text{orig}})$                  ▷ encode original prompt

  $E_{\text{cond}}^{\text{edit}} \leftarrow E_P(P_{\text{edit}})$                  ▷ encode edited prompt

  **for** $t = 1$ *to* $T$ **do**

       $A_{\text{orig}} \leftarrow \text{Get\_Attention\_Maps}(z'_{t-1}, E_{\text{cond}}^{\text{orig}})$

       $A_{\text{edit}} \leftarrow \text{Get\_Attention\_Maps}(z'_{t-1}, E_{\text{cond}}^{\text{edit}})$

       $z'_t \leftarrow \text{Forward\_Step}(z'_{t-1}, E_{\text{cond}}^{\text{edit}}, E_{\text{uncond}}, A_{\text{orig}}, A_{\text{edit}})$

                        ▷ Forward diffusion with attention control

  **end**

  $I_{\text{CF}} \leftarrow \text{Decode}(z'_T)$

**Step 3: Evaluate Edit Quality**

  $S_{\text{CLIP}} \leftarrow \text{Evaluate\_CLIP}(I_{\text{orig}}, I_{\text{CF}}, P_{\text{orig}}, P_{\text{edit}})$       ▷ CLIP similarity score

**return** $I_{CF}$

---

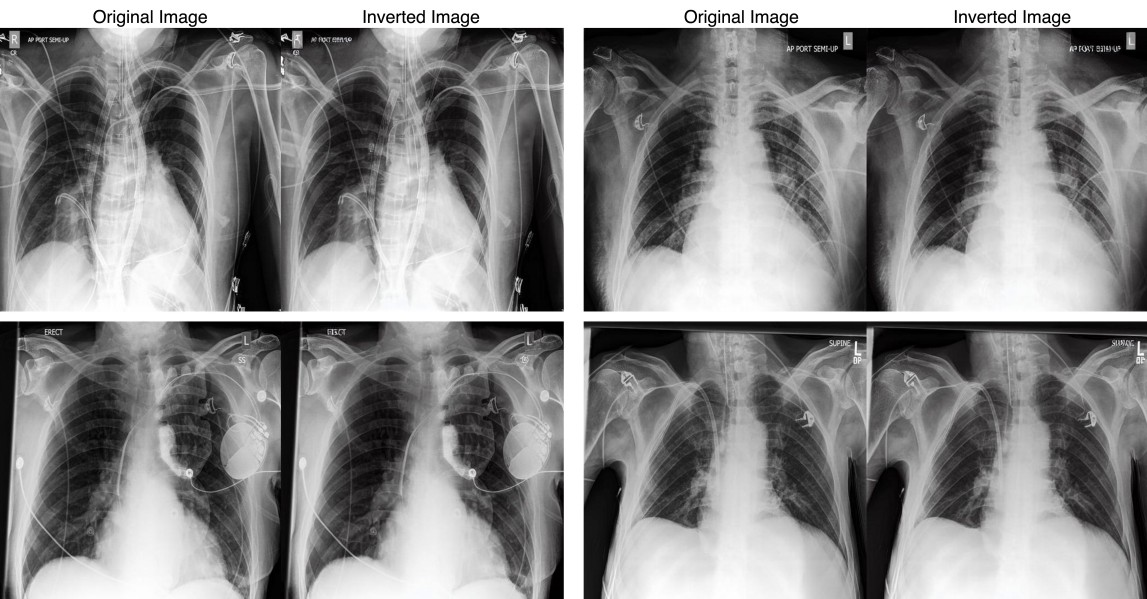

Figure 5: The inversion quality of the proposed generative model.

## Appendix B. Qualitative Comparisons to Image Editing Methods utilizing a Stable-Diffusion Backbone

This section presents additional qualitative comparisons to other language-guided image-editing methods that use a stable-diffusion backbone, namely Imagic (Kawar et al., 2023)[1], Null-text inversion (Mokady et al., 2023)[2], and RadEdit (Pérez-García et al., 2025) [3] **RadEdit** uses Stable Diffusion models fine-tuned to multiple chest x-ray datasets such as CheXpert, MIMIC-CXR and ChestX-ray 8 along with . The method employs two masks: an edit mask indicating the area where changes should be applied based on a text prompt and a keep mask that ensures other critical regions remain unchanged. These masks are combined with classifier-free guidance to ensure that edits are localized and consistent. RadEdit is trained on approximately 487k chest radiographs (compared to PRISM, which is trained on 80k images).

**Imagic** follows a three-step approach for language-guided image editing: (i) text embedding optimization to generate images similar to the input image based on the target text; (ii) generative model fine-tuning to improve the fidelity to the input image while freezing the optimized embeddings; and (iii) linear interpolation between the target text embedding and the optimized embedding and then, the generative diffusion process manipulates this combined representation to generate the final edited counterfactual (CF) image.

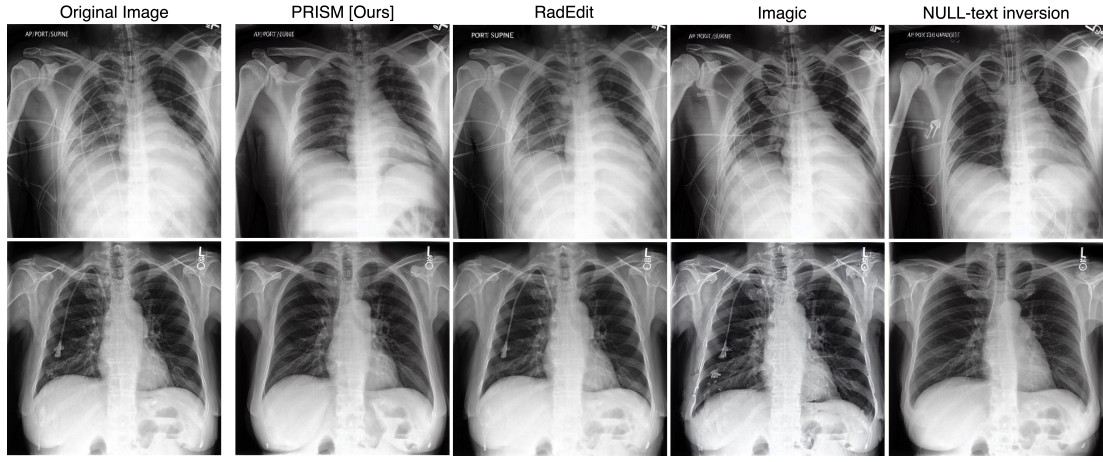

Figure 6: Comparison between PRISM (our method), RadEdit (Pérez-García et al., 2025), Imagic (Kawar et al., 2023) and Null-text inversion (Mokady et al., 2023) for the task of removing support devices from the original image. The edit text for PRISM, Imagic and Null-text Inversio was `Chest x-ray of a subject without support devices` while for RadEdit it was `remove support devices`. Note that RadEdit and Imagic is unable to remove support devices from the given image while Null-text inversion changes the patient's attributes. PRISM, Imagic and Null-text Inversion also use the same fine-tuned Stable Diffusion for image editing, while RadEdit uses their publicly released weights.

---

1. Since the original implementation is unavailable for Imagic, we use the code available at https://github.com/ShivamShrirao/diffusers/blob/main/examples/imagic/train_imagic.py
2. Source code for Null-text inversion is available at https://github.com/google/prompt-to-prompt
3. Source code for RadEdit is available at https://huggingface.co/microsoft/radedit.

**Null-text inversion**, uses DDIM inversion to map the input image to a sequence of noised latent codes that serve as pivotal latent codes, a reference point for further optimization. Next, the classifier-free guidance involves predicting noise twice: once conditionally with a text prompt and once unconditionally (using a null-text embedding). By optimizing around the pivotal latent codes, the null-text embedding is adjusted to align with the pivotal codes, allowing for efficient and high-fidelity editing of images using text prompts.

Fig. 6 show PRISM performs significantly better than RadEdit, Imagic, and Null-text inversions for removing devices from the original image. It should be noted that the methods Imagic and Null-text inversion were originally deployed with Stable Diffusion 1.4. For a fair comparison to PRISM, these two architecture use the same fine-tuned model as the PRISM for synthesizing images in Fig. 6.

### B.1. Sequential Image Editing

The image-editing performance of PRISM was evaluated against RadEdit (Pérez-García et al., 2025) in sequential image-editing scenarios. Fig. 7 presents a comparative demonstration where both methods were tasked with first adding and then removing a medical support device from an image. While RadEdit successfully added medical devices to the image, it shows limitations when attempting to remove these same devices.

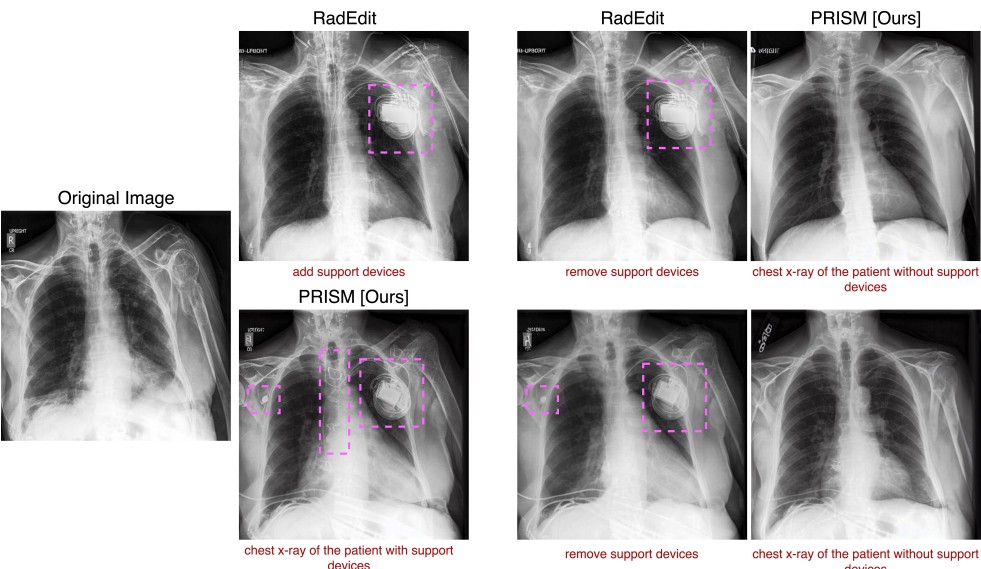

Figure 7: **Sequential editing comparison:** RadEdit (top) and PRISM (bottom) first add a support device to the original image. When prompted to remove these support devices, RadEdit fails or only partially succeeds, while PRISM successfully removes them completely. Note that RadEdit operated without masks in all experiments.

## Appendix C. Application of PRISM on ISIC Dataset

We extend the applicability of PRISM to a different imaging modality to show its effectiveness. We use the ISIC 2019 dataset (Tschandl et al., 2018; Codella et al., 2018; Combalia et al., 2019), a large-scale collection of dermoscopic images for skin cancer detection and classification. The 2019 version of the dataset contains 25,331 dermoscopic images across 8 different categories such as Melanoma (MEL), Melanocytic nevus (NV), Basal cell carcinoma (BCC), Actinic keratosis (AK), Benign keratosis (BKL), Dermatofibroma (DF), Vascular lesion (VASC), Squamous cell carcinoma (SCC). These dermoscopic images also contain artifacts such as dark corners, hairs, gel bubbles, rulers, ink, and patches.

As done for the CheXpert data in this manuscript, the tabular information is converted to sentences using the template `a dermoscopic image with [disease] showing [artifacts]` (Fig. 8). Due to the limited availability of the number of samples across different skin cancer types, we consider MEL and NV only as the `disease` types; and hairs, gel bubbles, rulers, and ink as the `artifacts`. Thus, the Stable Diffusion v1.5 is trained on 12,000 dermoscopic images for 50 epochs.

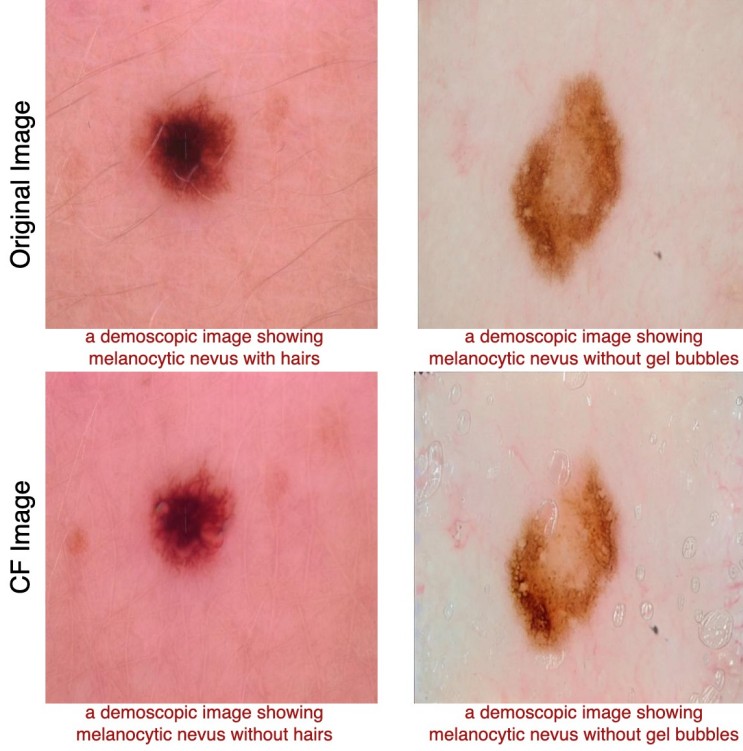

Figure 8: Deploying PRISM to remove/add artifacts on the demoscopic images in ISIC data. The corresponding text prompts are available at the bottom of the image. Note the selective removal of hair (left) or addition of gel bubbles (right) to the factual images.

## Appendix D. Classifier Performance on the Synthesized CF Images

We use the classifier, Efficient-Net, in Table 3 to validate the changes made when synthesizing CF images. Classifications across all heads of the classifier, along with the corresponding original and counterfactual images, are presented in Fig. 9. As shown, the intervened-upon attribute is successfully pushed across the decision boundary, while all other attributes retain their original classification. Notably, even when multiple attributes are present in the original image, only the targeted attribute undergoes a shift across the decision boundary, which is verified by the resulting counterfactual image. This demonstrates our model's ability to precisely distinguish and modify each attribute as intended.

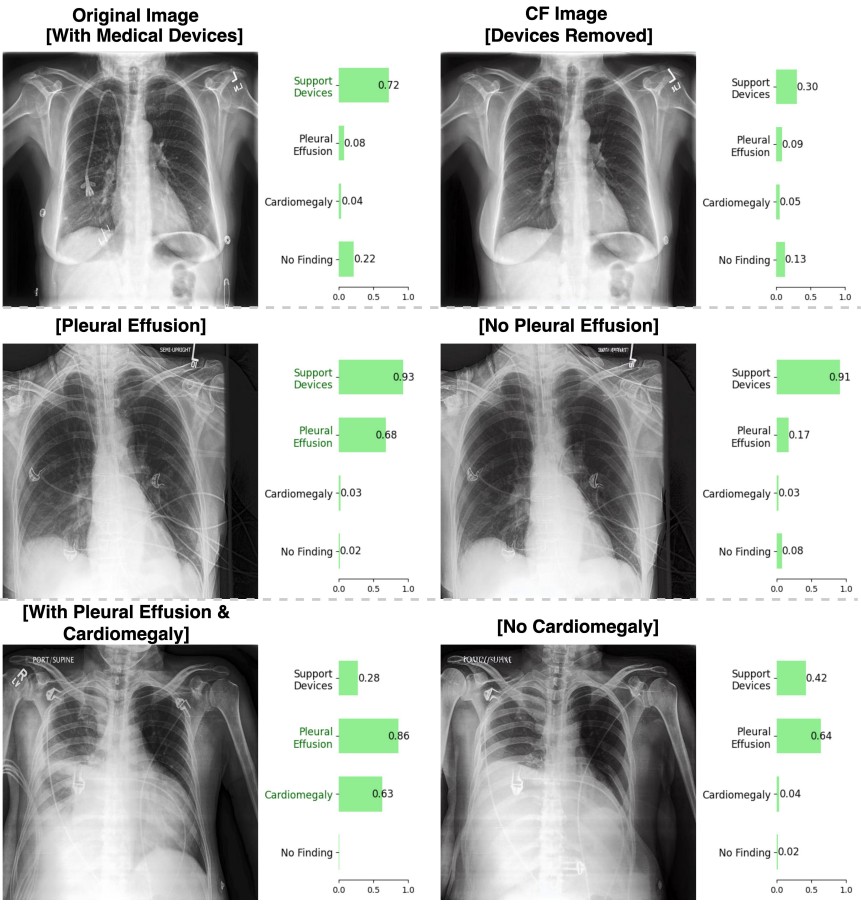

Figure 9: Classifier's performance on the original (left) and CF images (right). Note that the classifier is robust to changes made in the CF image. Text indicated in green shows the ground truth for the given image.

## Appendix E. Performance of the robust classifier

To evaluate the utility of counterfactuals synthesized from PRISM for downstream tasks, we augment our dataset and retrain the original EfficientNet multi-head classifier (see Table 3). Notably, the original classifier, trained without augmented counterfactuals, continues to detect support devices even after their removal—likely due to the correlation between pleural effusion and medical devices in the dataset. By incorporating CF augmentation, the classifier learns the true features associated with the medical device, reducing its reliance on correlations with the disease, see Fig. 10.

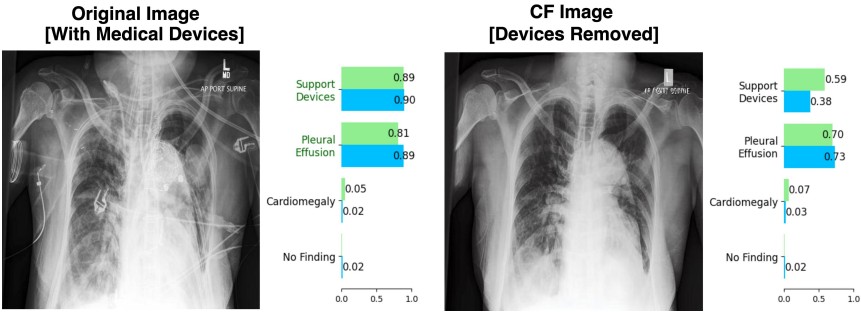

Figure 10: Original classifier and robust classifier performance on the original (left) and CF images (right) for the text prompt to remove medical devices. Note that the robust classifier is able to correctly identify the absence of medical devices in the CF image while the original classifier fails. Text indicated in green in the image shows the ground truth for the given image.

## Appendix F.  Validation of Image Modification Using State-of-the-Art VQA Models

To verify that the image modifications reflect expected anatomical changes, we use state-of-the-art VQA models to classify the images and analyze the disease-related features. We applied two state-of-the-art Vision Question Answering (VQA) models - Claude 3.5 Sonnet and LlaVA-Med. These models were chosen as they achieved high performance in disease diagnosis (Kurokawa et al., 2024). During inference, we input an image $X$ and its corresponding edited image $X'$ after modifying the text $T$: "an x-ray with severe pleural effusion" to $T'$:"an x-ray with no pleural effusion". The resulting correspondence with the VQA models is shown in Table 4. The response from the VQA models indicate not only a successful shift in classification but also correct reasoning, aligning with the desired modifications.

| | Question | Does this image have pleural effusion? |
|---|---|---|
| 
Original Image | Claude 3.5 Sonnet | Yes, this chest X-ray shows evidence of pleural effusion.  There appears to be blunting of the right costophrenic angle and increased opacity in the lower right hemithorax, which are characteristic radiographic signs of pleural effusion.  The effusion appears to be unilateral (on the right side) and moderate in size based on this frontal (PA/AP) view. |
| | LLaVA-Med | Yes, the image shows bilateral pleural effusion, which means there is an accumulation of fluid in the pleural space on both sides of the lungs. |
| | Question | Does this image have pleural effusion? |
| 
CF Image
**[Ours]** | Claude 3.5 Sonnet | This chest X-ray appears to be normal without evidence of pleural effusion.  The costophrenic angles (the corners where the ribs meet the diaphragm) are sharp and clear bilaterally, which is a key finding that suggests absence of pleural effusion.  The lung fields appear symmetrical with normal transparency, and there is no blunting of the costophrenic angles or visible fluid levels that would indicate pleural effusion. |
| | LLaVA-Med | No, the chest X-ray image does not show any pleural effusion. |

Table 4: Validation of the original and edited image using state-of-the art VQA models.

# Appendix G. Role of Confounders in CF Generation

An interesting correlation we observed when generating counterfactuals to explain the disease cardiomegaly (an enlarged heart) was the removal of the pacemaker. Note that a pacemaker can be a treatment for cardiomegaly. Indeed, when removing cardiomegaly, the CF showed that the pacemaker was also removed (Fig. 11), and other devices, such as EKG (electrocardiogram) stickers and tubes, are maintained (see final row of Fig. 9 and the bottom left example in Fig. 11). In the context where the original image contains both pleural effusion and cardiomegaly, the CF image with the task of removing cardiomegaly also removes the pacemaker. However, when generating a CF image to remove pleural effusion, no such effect occurs (Fig. 12). This suggests that the model associates the presence of a pacemaker specifically with cardiomegaly but not with pleural effusion.

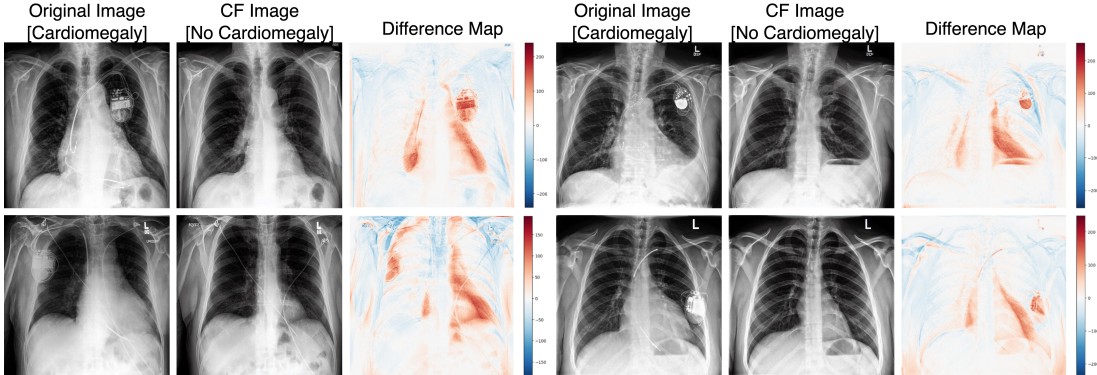

Figure 11: Samples where the removal of cardiomegaly, from the original image containing 'pacemaker'. Please note that our method removes the disease, cardiomegaly, and pacemaker.

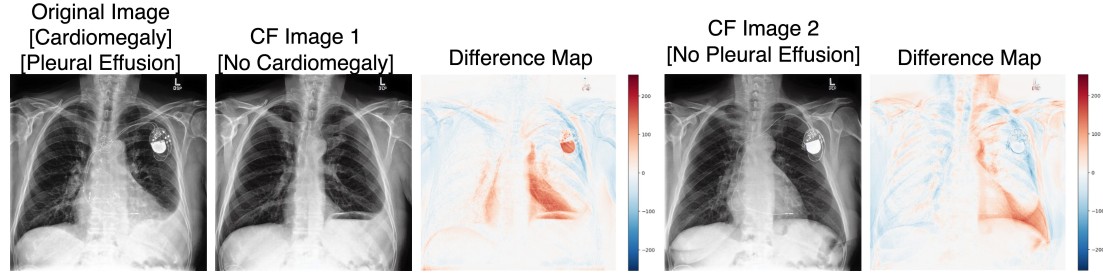

Figure 12: Comparing the change from original image with both cardiomegaly and pleural effusion to two different CFs. Note that when synthesizing the CF image with no pleural effusion the pacemaker is retained.

## Appendix H. Validation: CF generation in Challenging Cases

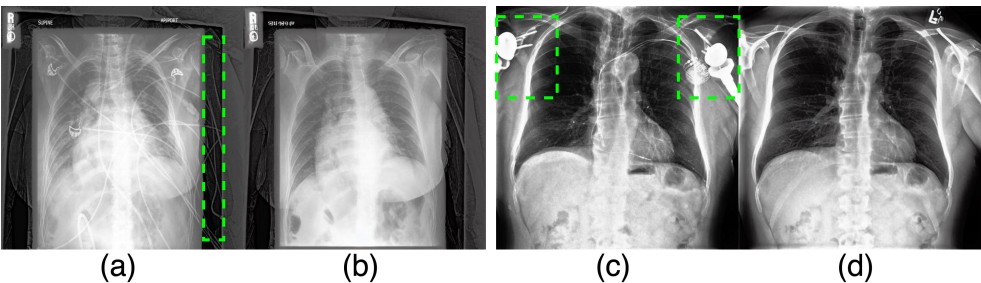

Figure 13: Samples showing challenging cases. (a,c): Original images with devices; (b,d): CF images without medical devices.

To demonstrate the robustness of PRISM, we examine cases that are particularly challenging to edit due to the placement of devices outside the field of view or devices in regions with bone structures. As shown in Fig. 13 (a), the device cables are located in low-light conditions near the arm. Fig. 13 (b) shows the edited image where the cables are accurately removed by our model without impacting the humerus. In Fig. 13 (c), the artificial shoulder joint creates high-intensity pixels. The corresponding edited image in Fig. 13 (d) shows the successful removal of the joints, replacing the affected pixels with feasible anatomical structures for the region. The structures in other areas are not altered. These examples demonstrate the robustness of the proposed method in challenging settings.

## Appendix I. Limitations of PRISM

Although our method is capable of synthesizing high-resolution images ($512 \times 512$), it faces difficulties in reproducing the small text written in the corner of radiographs (Fig 14) in both the inverted and CF images. This inability of Stable Diffusion to resolve fine text is a known phenomena and is also seen in natural images (Mokady et al., 2023).

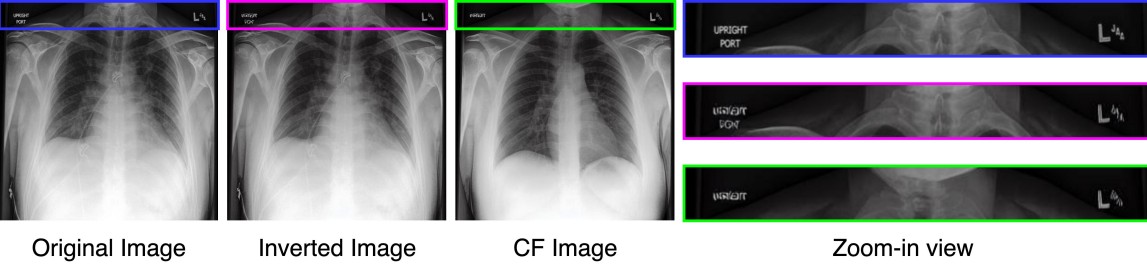

Original Image    Inverted Image    CF Image    Zoom-in view

Figure 14: Text at the corner of the image remains unresolved in the inverted and edited images.

These are challenging settings in which the model struggles to maintain consistent edits. This variation is partly dependent on the complexity of the image. For example, if there is significant overlap between the support devices and the anatomical features such as bone (as in Fig. 15 (c), the model attempts to remove the device and create regions that change the identity of the subject. In cases where the original image is distorted, the CF image deviates from expected changes (see Fig. 15).

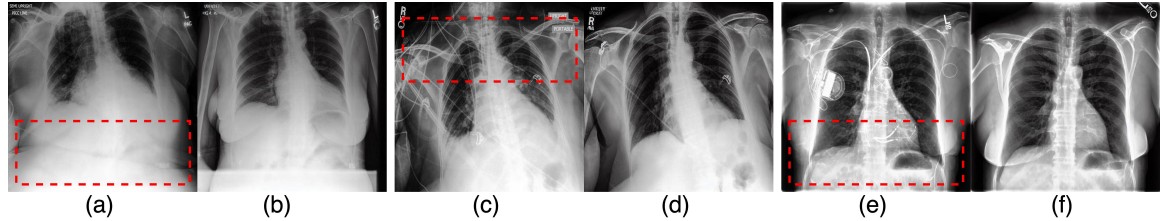

(a)    (b)    (c)    (d)    (e)    (f)

Figure 15: Examples of original (a, c, e) and CF image (b, d, f) pairs. The command was to remove the support device, and the edits were inconsistent with the expected outcome. Red boxes highlight areas where the changes are not as intended. (a-b): The radiograph shows a problem with the original image (at the bottom). The edited image incorrectly modifies this region instead of retaining it. (c-d): The red-boxed region contains multiple tubes. While removing the tubes, the model recreates the missing anatomical area improperly. (e-f): When removing the medical devices, the subject is depicted more strongly as female

