# OpenReview forum: "PRISM: High-Resolution & Precise Counterfactual Medical Image Generation using Language-guided Stable Diffusion"
_MIDL.io/2025/Conference — MIDL 2025 Oral_

### Official Review · Reviewer_tQdL · 2025-02-13

**Confidence:** 4
**Preliminary Rating:** 2
**Recommendation:** Poster
**Final Rating:** 4

**Summary:**

The authors propose to use factual and counterfactual images as a data-centric approach to improve classifier robustness. Their approach leverages inverting the image generation process and uses manually crafted negative prompts to generate counterfactual images. The demonstrated approach enhances the performance of a classifier on the downstream task of classifying diseases.

**Strengths:**

Improvements on downstream task demonstrate the feasibility of the proposed approach.

The presentation, especially the figures, are nice and give a good overview of the approach.

Extensive ablations and additional experiments in the supplements help deepen the understanding and take-home messages of the paper.

**Weaknesses:**

1. Missing Comparison to Other Methods: My main concern is that you are only comparing your results against a single other - gan-based - approach. Ever since SDv1.5 came out many people have experimented with the idea of using diffusion models to edit images. Examples are:
 - Pérez-García et al. 2024: RadEdit: stress-testing biomedical vision models via diffusion image editing
 - Kawar et al. 2023: Imagic: Text-Based Real Image Editing With Diffusion Models
 - Mokady et al. 2023: Null-text Inversion for Editing Real Images using Guided Diffusion Models
I believe the authors should at least acknowledge and ideally compare their approach to these publications.

2. Missing technical details: I believe the fine-tuning and sampling part should be more in the focus. Also, you don't mention how you perform image inversion.

3. In Appx F, you show how the Diffusion model learns the correlation between pacemaker and Cardiomegaly. In Appx D you claim the CF augmentation reduces the reliance on correlations. I believe these two claims are contradicting each other.

4. The writing in some parts is too lengthy and confusing. One example is in 3.2, where you mention that you train AlexNet to detect the presence of support devices, but this is only the case for this one specific experiment, I assume.

5. If I understand your experiment in Table 3 correctly, it uses twice the amount of data. Some people use diffusion models to augment the training dataset and increase the training corpus. This makes me question whether the improvements stem from counterfactuals or data augmentation.

**Detailed Comments:**

All my comments are mentioned in the weaknesses.

**Justification Of The Final Rating:**

Thanks to the authors for providing their detailed rebuttal with multiple experiments demonstrating the method's effectiveness. I believe that this addresses most of my concerns and therefore I am happy to raise my score. Best of luck.

**Justification Of The Preliminary Rating:**

At the current time I believe the paper lacks comparison to concurrent and recent work from other conferences. Additionally, missing technical details make it hard to understand the paper. Overall I give it a weak reject.

**Questions To Address In The Rebuttal:**

If you can explain why your method is unique and cannot be compared to the previous work I mentioned and improve the documentation of the technical details I consider increasing my score.

**Special Issue:**

No

---

> ### Author Response · Authors · 2025-03-08
>
> We sincerely appreciate the reviewer’s valuable and detailed feedback. We are pleased that our extensive ablations and additional experiments effectively conveyed our key messages. Below, we address each of the concerns raised:
>
> **[Comparison to Other Methods]**:
>
> We wish to reiterate that the claim of our work is not that we are the first to ever introduce a language-guided Stable Diffusion (SD) based image editing model. This work does, however, present the first language-guided, fine-tuned Stable-Diffusion model that enables high-resolution, high fidelity, counterfactual medical image generation and presents the code for the community to use for downstream tasks. While language-guided SD models have been explored in natural images, medical imaging poses unique challenges (e.g., spurious correlations like support devices in chest X-rays) that previous methods struggled to address and which we address in this work. We focused on illustrating our method’s potential, particularly in comparison to the current SOTA within medical imaging. We acknowledge the reviewer’s concern regarding the limited number of comparisons, especially with models using the Stable Diffusion backbone. In order to address this concern, we added a new section in **Appendix B** that provides a detailed comparison of PRISM with the methods mainly RadEdit [Pérez-García et al. 2024], Imagic and Null-text inversions. For fair comparisons, we use the same fine-tuned model for image editing as the PRISM for Imagic and Null-text inversions. We observe that RadEdit and Imagic cannot remove medical devices when prompted to do so successfully. Null-text inversion removes the devices in some cases but also alters the anatomy, such as the rib-cage structure, of the subject. We would like to highlight that a comparison to RadEdit would be unfair as the denoising U-Net is trained on approximately 487K images (compared to our 80k images). Additionally, RadEdit uses two masks - edit and keep mask, to make edits only in the localized regions, whereas PRISM does not require masks. However, we still compare to RadEdit using the pre-trained weights initially provided by the authors. We also acknowledge these works in our main manuscript.
>
> **[Missing Technical Details]**: We thank the reviewer and have added additional technical details to improve the clarity. We have expanded the main manuscript to provide a more detailed explanation of our fine-tuning and sampling procedures in Section 3.1. Additionally, a dedicated section in Appendix A further elaborates on these details.
>
> **[Addressing the Correlation Between Pacemakers & Cardiomegaly]**: The reviewer points out an important correlation between pacemaker and cardiomegaly (*Appendix E* and *Appendix G*). Originally, we added these figures to the appendix as we found them to be interesting as the model seems to uncover a relationship between attributes (e.g. pacemaker and disease),  when no specific information about pacemakers, or causal relationships, was provided. According to the medical literature [Fung et. al 2007: “The Pacing to Avoid Cardiac Enlargement(PACE) trial:clinical background,rationale,design and implementation”], pacemakers are often used to cure cardiomegaly for some subgroups of patients. In some cases, pacemakers cause cardiomegaly [Koo et al. 2020 “Pacing-induced Cardiomyopathy”]. This indicates that there is a possible correlation/causation between the attributes that the model is uncovering. The model learns these correlations/causations internally. This is the subject of further investigation.
>
> **[Text Refinement & Addressing Lengthy Areas]**: We have revised *Section 3.2* to include details about how many samples were used for the augmentation experiments and the purpose of the two classifiers. Additionally, we have refined the text, in several places (including the appendix). We welcome any further suggestions on areas that could benefit from additional clarification.
>
> **[Augmenting Sampled Images vs CF]**: In order to address the reviewer’s concern regarding whether improvements stem from counterfactuals or general data augmentation. We conducted an additional experiment, now included in **Table 3** (below). We explicitly compare using PRISM-generated CFs against standard augmentation techniques and examine the improvement in the downstream task. Our results confirm that the observed performance gains are due to CF augmentation rather than simple data expansion.
> |  | Pleural Effusion | Cardiomegaly | No Finding | Support Devices
> |-|-|-|-|-
> | Original Data | 0.80 | 0.87 | 0.91 | 0.86
> | Original Data + SD samples | 0.82 | 0.86 | 0.91 | 0.85
> | Original Data + PRISM CFs | **0.88** | **0.90** | **0.92** | **0.88**
>
> We appreciate the reviewer’s thoughtful insights and believe our revisions address the concerns raised. We welcome any additional questions and look forward to further engaging on improving this work. Thank you again for your time and effort in reviewing our work.

---

### Official Review · Reviewer_dAxz · 2025-02-17

**Confidence:** 4
**Preliminary Rating:** 5
**Recommendation:** Oral
**Final Rating:** 5

**Summary:**

This paper introduces a method called PRISM for high-resolution and precise counterfactual medical image generation, leveraging a fine-tuned Stable Diffusion model guided by text prompts. The approach enables selective editing of disease attributes and medical devices (e.g., pacemakers) in chest X-ray images with minimal disruption to surrounding anatomy. Through extensive experiments on the CheXpert dataset, the authors demonstrate that PRISM improves both visual fidelity and class-consistency over baseline methods, while also enhancing downstream classification tasks by mitigating dataset biases and spurious correlations.

**Strengths:**

## Strength

- The approach supports 512×512 resolution, providing detailed and clinically relevant images that surpass many existing GAN-based counterfactual methods.

- By harnessing CLIP-based text encodings, the method enables intuitive, targeted modifications (e.g., removing devices or disease features) through simple natural language prompts.

- Freezing most parts of the Stable Diffusion architecture and only refining the U-Net component offers a balanced approach, leveraging general pre-trained features while keeping computational costs feasible.

- Incorporating generated counterfactual images into model training can reduce spurious correlations (e.g., devices vs. disease) and lead to more robust downstream classification.

The paper is well-structured, with clear illustrations (e.g., figures on system architecture and use cases) that make the methodology easy to follow.

**Weaknesses:**

## Weakness

- The method is primarily demonstrated on chest X-ray data from CheXpert, leaving its effectiveness on other imaging modalities and more diverse clinical contexts unclear.

- While the paper compares against a GAN-based method, further experiments or ablations (e.g., evaluating different fine-tuning configurations or additional baselines) could strengthen the evidence for the proposed approach.

- The model’s clinical relevance remains partially unverified, as evaluations mainly rely on automated metrics without extensive expert review or real-world testing.

- In scenarios with multiple overlapping devices or extreme imaging artifacts, the generated counterfactuals may exhibit partial edits, residual artifacts, or anatomically implausible features.

**Detailed Comments:**

**1) Identity Preservation**
The paper uses L1 distance or feature-space distance to measure if only the specified attribute changes, but metrics alone may be insufficient. Are there organ deformations or skeletal misalignments after removing devices or lesions? Overlaying original and counterfactual images at key landmarks (e.g., ribs, sternum) could confirm that edits remain localized.

**2) Multiple Lesions/Devices**
Real clinical scenarios often have multiple diseases or devices in a single X-ray. If the paper only shows single-attribute edits, it may not fully demonstrate robustness. Showing examples with multiple lesions or multi-step edits (e.g., removing cardiomegaly, then pleural effusion) can highlight whether repeated edits introduce artifacts.

**3) Failure Cases and Visualization**
The paper notes potential issues with small text markers (e.g., L/R) or noisy corners but lacks detailed failure examples. Providing more such cases in an appendix, along with brief root-cause analysis, would help clarify the method’s limitations.

**Justification Of The Final Rating:**

Thank you for your thorough and detailed response to my earlier concerns. I appreciate how comprehensively you've addressed each point, particularly regarding the additional experiments with different imaging modalities and the detailed analysis of failure cases. Your explanations and the supplementary materials you've added have effectively clarified my questions. I'm satisfied that all my concerns have been adequately addressed in this rebuttal.

**Justification Of The Preliminary Rating:**

I strongly recommend acceptance. While further exploration—such as testing across more imaging modalities or handling multiple lesions—is necessary, the paper’s approach to high-resolution counterfactual generation is timely and impactful. Although the engineering effort may be significant, the fine-tuning strategy, language-guided edits, and improved classification performance show clear clinical potential, making this a valuable contribution.

**Questions To Address In The Rebuttal:**

Same above.

**Special Issue:**

No

---

> ### Author Response · Authors · 2025-03-08
>
> We sincerely appreciate the reviewer’s thoughtful and detailed feedback, as well as their strong recommendation for acceptance. We are pleased that they find our paper clear and well-structured, and we are thrilled that our approach is considered timely and impactful. Below, we first address the reviewer’s key suggestions, and then outline some additions we made to the paper based on the comments.
>
> **Weaknesses**
> --------------------
>
> **[Additional imaging modalities/diverse clinical contexts]**: The reviewer suggested experimenting with additional imaging modalities or clinical contexts. In response, we ran additional experiments on dermoscopic images to demonstrate the method’s versatility across different datasets. We mention this additional experimentation in the main text and show results in **Appendix C**.
>
> **[Additional comparisons and experiments]**: The reviewer mentions running additional comparisons to baselines. We do so in Appendix B, where we compare our method to others such as RadEdit [Pérez-García et al. 2024], Imagic [Kawar et al. 2023] and Null Text Inversions [Mokady et al. 2023] that use a Stable Diffusion backbone.
>
> **[Clinical relevance?]**: We agree with the reviewer that the clinical relevance of our model requires extensive evaluation by clinical experts before full validation and may aid us in future developments of this work.
>
> **[Counterfactuals under extreme artifacts?]**:  This is addressed below.
>
> **Detailed Comments**:
> ----------------------------
>
> **[Do other key landmarks change beyond the specified attribute change?]**  We acknowledge the importance of ensuring that counterfactual edits maintain anatomical plausibility throughout the image. However, an in-depth investigation to uncover errors across key (established) landmarks when specified attributes are changed is important; in complex medical imaging contexts, such as the ones investigated in this work, this presents enormous challenges, especially given that the required image labels are not provided across the structures in the images. Given that this is the first paper on this topic, we showed how **difference maps** (pixel-wise differences between the original and edited images) highlight the regions affected by the modifications. Our results indicated that the model focused on the attribute of interest with little effect on the rest of the image. We have added visualizations to the work, overlaying factual and counterfactual images, along with the difference maps, and added these to the supplementary materials. These animations are also available at the project website https://amarkr1.github.io/PRISM/#results
>
> **[Multiple Diseases/Devices]**: We would like to first emphasize that the CheXpert dataset does not contain lesions but does include multiple diseases and medical devices within images. As demonstrated in **Appendix G Figure 12**, our model can change the part of the image associated with the disease with high fidelity. The reviewer raises an interesting point about sequential attribute removal. In response, we have conducted an ablation study evaluating the successive image edits, shown in **Appendix B.1 Figure 7**. We would happily include more of these results in an additional appendix section if required.
>
> **[Failure Cases and Visualization]**: As detailed in **Appendix I**, we discuss common failure cases, particularly related to text inconsistencies when inverting and editing images. During a root-cause analysis, we noted that these issues are inherent to Stable Diffusion itself, which struggles with text generation within medical images [Mokady et al. 2023: Null-text Inversion for Editing Real Images using Guided Diffusion Models]. However, addressing this limitation is beyond the scope of this particular paper.
>
> That being said we added a set of examples of true failure cases in **Appendix I, Figure 15**. These cases showed:
>
> * **Inconsistent device removal**: The model does not fully remove a support device or unintentionally modifies surrounding regions in a few challenging cases, as mentioned by the reviewer.
> * **Inaccurate anatomical reconstruction**: When removing multiple tubes, the model struggles to recreate the occluded anatomical structure in a few cases.
> * **Unintended identity shifts**: In a few cases, the removal of medical devices causes the subject's appearance to shift in a way that alters perceived demographic attributes (e.g., making the individual appear more female).
>
> We provide some additional hypotheses as to what could be the potential root causes for these cases, as well as insights on how to overcome some of these challenges.
>
> We hope these additions strengthen our paper and provide clarity regarding the model’s strengths and limitations. Once again, we greatly appreciate the reviewer’s time and valuable feedback.

---

### Official Review · Reviewer_grY7 · 2025-02-19

**Confidence:** 5
**Preliminary Rating:** 5
**Recommendation:** Oral
**Final Rating:** 5

**Summary:**

This paper proposes a counterfactual framework by leveraging the pre-trained high-resolution image generative model (SD1.5), with tabular data conversion and lower noise scheduler design for finetuning SD.

**Strengths:**

1. The motivation is clear, and the text description is clarified.
2. The finetuning SD and tabular data conversion for medical counterfactual synthesis is relatively novel, and the quantitative evaluation is meaningful.
3. The experimental results are sufficient, promising, and clinically highly related.

**Weaknesses:**

Overall, I appreciate this paper, but the technical novelties can be improved more efficiently, the proposed method can be evaluated in more 2D medical modalities in the following work, and some implementational functions, e.g. lower noise scheduler, can be further strengthened.

**Detailed Comments:**

Overall, I appreciate this paper, but the technical contributions and experimental sufficient can be improved in the following ways:

1. Technical Improvements: This medical counterfactual synthesis is akin to personalized synthesis in computer vision. Fine-tuning all parameters of the SD UNet is not ideal. The authors could explore using ControlNet or LORA fine-tuning for the attention layers for further enhancement. Additionally, leveraging more advanced autoregressive visual language models, like Janus-Pro, could lead to technical improvements.

2. Modality Usage: The method could be extended to other 2D modalities, such as skin images, OCT, etc., to increase its applicability.

3. Synthetic Fidelity: The high synthetic fidelity in unaffected or condition-related regions may be due to using a lower noise scheduler, rather than DDIM alone. The authors should clarify this point and acknowledge that while a lower noise scheduler harms diversity in condition-related regions, it also accelerates fine-tuning time.

**Justification Of The Final Rating:**

The rebuttal is effective. The focused problem and corresponding experiments are sufficient and achieve SOTA performance. Papers that align well with the conference's scope are recommended for ORAL acceptance.

**Justification Of The Preliminary Rating:**

The proposed method is novel, using the impressive generative power of the larger synthesis model. The description is clear, and the experimental evaluation is thorough and sufficient. Therefore, I recommend accepting this paper.

**Questions To Address In The Rebuttal:**

I look forward to the authors releasing the corresponding code, model weights, and details on data preprocessing and splitting.

**Special Issue:**

Yes

---

> ### Author Response · Authors · 2025-03-08
>
> We would like to sincerely thank the reviewer for their very positive feedback on our paper. We are pleased to hear that they find our work promising and highly clinically relevant.  As mentioned in our manuscript, we will release the well-documented code and model weights upon acceptance of the paper.
>
> The reviewer brought up a few suggestions for future improvement of work, mentioning some possible technical improvements and additional experiments. We added a few of these suggested experiments to the paper.
>
> **[Additional contexts/modalities]**: The reviewer suggested extending the method to other 2D contexts. We ran additional experiments on dermoscopic images from the ISIC dataset to demonstrate its versatility and clinical relevance.  We accepted these suggestions and included them in the main text and reference  **Appendix C** for more details.
>
> **[Synthetic fidelity]**: We agree with the reviewer that there is a trade-off between identity preservation and diversity associated with the value of noise scheduler. As suggested by the reviewer, we added additional details and provided some insights about this in section 3.1, such as:
>
> *“PRISM uses the default DDPM scheduler for fine-tuning the model `runwayml/stable-diffusion-v1-5'. There is a known tradeoff between a lower noise scheduler and diversity in the sampled results. Choosing a lower noise scheduler (12e-3 in this case) tends to produce a more detailed image with less noise and generates deterministic results. Additionally, the convergence is faster, as only a few sampling steps are required. The implications of this tradeoff should be explored in each context of interest. ”*
>
> Given the limited time for the rebuttal, we were not able to explore the tradeoff in several contexts but will include this in future work.
>
> **[Technical improvements on fine-tuning SD]**: The reviewer suggested possibly exploring methods such as ControlNet or LoRA as alternate strategies to fine-tune the Stable Diffusion architecture. While the primary focus of this first paper was not on optimizing the fine-tuning strategies, this will be the focus of future work.

---

> > ### Comment · Reviewer_grY7 · 2025-03-08
> > **Effective Rebuttal Justifies Acceptance**
> >
> > The rebuttal is effective. While I acknowledge that the technical novelty could be further improved, the focused problem and corresponding experiments are sufficient and achieve SOTA performance. Overall, I believe this paper should be accepted. Papers that align well with the conference's scope are recommended for ORAL acceptance.

---

### Author Response · Authors · 2025-03-08

We thank the reviewers for their thorough and constructive feedback on our manuscript. We appreciate the overall positive reception of our work, especially with regards to its novelty within medical imaging, strong experimental evaluation, and clinical relevance. Based on the reviewers' helpful comments and suggestions, we have incorporated additional baseline comparisons, ran experiments on an additional 2D dataset (dermoscopic images from the ISIC dataset), provided more ablations, and further refined our analysis in the paper. We will release the code once the paper is accepted, and ensure that our code is well-documented and that the dataset usage details are publicly available, in order to support reproducibility and further research in this area. We have submitted *.gif files as supplementary materials to address the reviewer's feedback regarding the visualisation of edits made to the original image. These animations can also be viewed on our project website: https://amarkr1.github.io/PRISM/#results.

Below, we address the specific concerns raised by each reviewer and outline the improvements made in the updated paper. All changes made to the main manuscript, including in the appendix, are highlighted in blue text. Note that due to the significant number of experiments (many added to the appendix), our complete manuscript is now 26 pages long.

---

### Author Rebuttal · Authors · 2025-03-08

**Rebuttal:**

Please find our revised manuscript (including appendix) along with supporting materials within the zip file. We include three *.gif files - Devices_to_NoDevice.gif, Pleural Effusion_to_Healthy.gif and Cardiomegaly_to_Healthy.gif as supporting materials to address the reviewer's feedback regarding the visualisation of edits made to the original image.

**Supporting Material:**

/attachment/be1c69f7e8ff8e315e101ba8e74b6c93e0cbe185.zip

---

### Meta-Review · Area_Chair_Uuv2 · 2025-03-19

**Recommendation:** Accept (Oral)
**Confidence:** 4

**Metareview:**

All reviewers found the proposed method to be novel and the results promising.